# Droplet Digital PCR Enhances Sensitivity of Canine Distemper Virus Detection

**DOI:** 10.3390/v16111720

**Published:** 2024-10-31

**Authors:** Victoria Iribarnegaray, Guillermo Godiño, Camila Larrañaga, Kanji Yamasaki, José Manuel Verdes, Rodrigo Puentes

**Affiliations:** 1Microbiology Unit, Department of Pathobiology, Faculty of Veterinary, Universidad de la República (Udelar), Route 8 Km 18, Montevideo 13000, Uruguay; victoria.iribarnegaray@pedeciba.edu.uy; 2Pathology Unit, Department of Pathobiology, Faculty of Veterinary, Universidad de la República (Udelar), Route 8 Km 18, Montevideo 13000, Uruguay; guille030599@gmail.com (G.G.); camilalarranaga@gmail.com (C.L.); yamasaki-kanji1914@outlook.jp (K.Y.); jose.verdes@fvet.edu.uy (J.M.V.)

**Keywords:** canine distemper virus, droplet digital PCR, molecular diagnosis

## Abstract

Canine distemper virus (CDV) poses a substantial threat to diverse carnivorans, leading to systemic and often fatal diseases. Accurate and prompt diagnosis is paramount for effective management and curbing further transmission. This study evaluates the diagnostic performance of droplet digital PCR (ddPCR) in comparison to conventional reverse-transcription (RT-PCR) and quantitative reverse-transcription real-time PCR (RT-qPCR). Seventy-six clinical samples were collected from dogs with CDV symptoms diagnosed by specialized veterinarians, and sixteen samples from apparently healthy individuals. Conventional PCR, quantitative real-time PCR, and ddPCR were deployed, and their diagnostic capabilities were meticulously assessed. DdPCR exhibited heightened analytical sensitivity, reaching a detection limit of 3 copies/μL, whereas RT-qPCR had a detection limit of 86 copies/μL. The comparative analysis between clinical diagnosis and molecular techniques, including RT-PCR and RT-qPCR, demonstrated low concordance, with Kappa coefficients of 0.268 and 0.324, respectively. In contrast, ddPCR showed a moderate concordance, with a Kappa coefficient of 0.477. The sensitivity was 42.4% for RT-PCR, 57.9% for RT-qPCR, and 72.4% for ddPCR, with 100% specificity for all methods. This study underscores ddPCR’s superior sensitivity and agreement with clinical CDV diagnosis, even at low viral concentrations, suggesting it as a promising alternative for CDV diagnosis.

## 1. Introduction

Canine distemper virus (CDV) belongs to the genus *Morbillivirus* within the family *Paramyxoviridae* and is an enveloped negative-polarity RNA virus [1]. *Morbilliviruses* are highly contagious pathogens that cause outbreaks of systemic, often fatal diseases in animals and humans worldwide. This genus also includes measles virus (MeV), Rinderpest virus (RPV), peste des petits ruminants virus (PPRV), phocine distemper virus (PDV), cetacean morbillivirus (CeMV), and the recently discovered feline morbillivirus (FeMV) [1].

CDV infects a wide range of carnivorans, including members of the families *Ailruidae*, *Canidae*, *Cercopithecidae*, *Felidae*, *Hyaenidae*, *Mustelidae*, *Procyonidae*, *Tayassuidae*, *Ursidae*, and *Viverridae* [2,3,4,5]. It exhibits the second highest fatality rate of infectious diseases in dogs, after rabies [6]. The disease caused by CDV is multi-systemic, affecting the respiratory, gastrointestinal, and nervous systems. Early stages of infection are marked by non-specific clinical signs such as anorexia, depression, conjunctivitis, and hyperkeratosis of digital cushions [7]. At this stage, viruses can be found in every secretion of the infected animal [8]. As the disease progresses, a more pronounced and severe clinical picture emerges, including catarrhal inflammation of the bronchi and larynx, along with episodes of vomiting and diarrhea. When the nervous system is affected, apathy, ataxia, and paralysis may ensue, progressing from paraplegia to quadriplegia [9,10]. Dogs with nervous system pathologies often do not survive or exhibit lifelong neurological signs [11]. The progression of canine distemper is often variable, wherein the manifestation of clinical signs is contingent upon the specific strain of the virus responsible for the infection. This variability of clinical signs depends on the age of the host, its immune status, and the virus strain, with more than 50% of cases being probably subclinical [6,12]. 

Rapid and accurate diagnosis of CDV disease facilitates the treatment and conservation of infected dogs and prevents further transmission to susceptible hosts. Therefore, it is important to choose a rapid and valid diagnostic method. Variable signs of distemper and subclinical status of the infection in some cases are the main challenges of diagnosis based on physical examination. Currently, routine laboratory and clinical findings are helpful for initial diagnosis but not sufficient for confirmation of the infection [12,13]. In recent years, advancements in diagnostic methodologies have introduced more sensitive and specific techniques for the accurate detection of CDV, such as reverse-transcriptase PCR (RT-PCR), quantitative reverse-transcription real-time PCR (RT-qPCR), enzyme-linked immunosorbent assay (ELISA), immunofluorescence assay (IFA), neutralization antibody (NA) test, and rapid immunochromatographic assays [14,15,16]. Antibody detection often proves limited in diagnosing CDV due to potential interference from previous vaccination and/or post-infection status. Consequently, diagnostic approaches based on antigen or nucleic acid detection hold greater value and timeliness [17]. Notably, rapid immunochromatographic (IC) antigen test kits, known for their speed, cost-effectiveness, and user-friendliness, alongside RT-PCR assays, recognized for their heightened sensitivity and specificity, emerge as more practical choices for CDV diagnosis [12,14]. Various challenges are inherent in the detection of CDV, involving the identification of the optimal tissue samples, early detection, and the detection of target molecules at low concentrations with high sensitivity. 

Due to technological advances, various PCR variants have been developed, among which digital droplet PCR (ddPCR) is a novel platform designed to provide greater sensitivity and precision for the detection and quantitation of nucleic acid target molecules [18,19]. This technology not only enables detection and allows direct quantification without the need for a standard curve in a sample, but is also highly sensitive, even surpassing qPCR [20,21,22]. Studies have demonstrated that ddPCR can acquire the nucleic acid detection sensitivity of low viral loads (10^−5^) [23]. The heightened sensitivity of the technique is partly attributed to the fragmentation of the sample into a set of nanodroplets, each representing an individual reaction in an oil emulsion. The ability to diagnose CDV even when the agent is excreted in low quantities, either due to the timing of sample collection or partial immunity of the animal, is crucial for accurate diagnosis and management of the disease. This work assessed the diagnostic performance of conventional RT-PCR, RT-qPCR, and ddPCR for CDV detection in diverse samples and compared the results with clinical diagnosis.

## 2. Materials and Methods

### 2.1. Samples Collection

In this study, 76 blood, urine, and nasal/ocular secretion samples from dogs exhibiting clinical signs consistent with CDV disease were included, and 16 negative control samples were acquired from healthy individuals without clinical signs of CDV disease and evidence of not having been exposed to the virus in the environment or with other pets with possible subclinical infection. The samples were collected between 2021 and 2023 at the Microbiology Laboratory Services, Faculty of Veterinary, Montevideo, Uruguay. Blood samples were collected from the jugular, cephalic, or lateral saphenous vein and placed into sterile EDTA tubes. Urine samples were collected via cystocentesis, and nasal or ocular secretions were taken with a sterile swab and stored in 200 µL DNA/RNA Shield (Zymo Research). Cerebellum (left hemisphere) samples were collected fresh and immediately frozen at necropsy. All samples were stored at −80 °C. The right hemisphere was fixed in 10% neutral buffered formalin and processed routinely for histologic examination. 

Clinical diagnosis of the dogs was performed by specialized veterinarians, taking into account the presence of some the following clinical signs: intermittent or persistent fever; mucous or purulent discharge from the eyes and nose that may be accompanied by difficulty breathing; vomiting and diarrhea leading to dehydration; loss of appetite resulting in weight loss; lethargy and depression with the dog showing a lack of energy and activity; thickening of the paw pads and nose often referred to as “hyperkeratosis”; and neurological clinical signs such as seizures, muscle twitches, partial or total paralysis, and abnormal behaviors. The clinical manifestations used as a reference for the clinical diagnosis of canine distemper virus have been previously described by numerous authors [7,11,24]. For canines euthanized due to severe illness, the brain was sectioned into two hemispheres. The right hemisphere was used for conventional hematoxylin and eosin (H&E) stain, Luxol Fast Blue (LFB) stain, and immunohistochemistry against CDV. For immunohistochemistry, to confirm the presence of CDV antigens, we employed a mouse anti-CDV monoclonal primary antibody (MCA 1893, dilution 1:250; BioRad, Hercules, CA, USA) and secondary antibody (Mouse-on-Canine HRP-Polymer, Biocare Medical, Pacheco, CA, USA); positive antigen–antibody reactions were observed by incubation with 3,3′-diaminobenzidine tetrahydrochloride (DAB). Histologic lesions were categorized as acute, subacute, or chronic. The lesions were characterized by vacuolation, gliosis, perivascular cuffing, and CDV-positive cells in acute stage; patchy demyelination, prominent perivascular cuffing, and CDV-positive cells in subacute stage; or increased neuronal degeneration and necrosis in chronic stage [25,26]. The left hemisphere was used for molecular biology techniques to confirm the presence of and quantify the viral load in the tissue.

The experimental protocols for the canine studies performed in this paper were approved by Animal Use Ethics Commission (CEUA) from the Universidad de la República Oriental del Uruguay (approval number CEUAFVET-1003/20, Exp. 111900-000900-20).

The Onderstepoort strain of canine distemper virus (CDV-Ond), originally isolated in 1940, was cultured in Vero cells stably expressing canine signaling lymphocytic activation molecules (Vero.DogSLAM cells) [27]. These cells were kindly provided by Dr. Yusuke Yanagi (Kyushu University, Japan). The virus was propagated and titrated according to standard protocols, and the viral stock was prepared for use in this study.

### 2.2. RNA Viral Isolation and Reverse Transcription

Viral RNA extraction from urine, nasal and ocular secretions, and blood was carried out using the RNA Viral Kit (Zymo Research, Irvine, CA, USA) following the manufacturer’s recommendations. For cerebellum samples, TRI Reagent (Zymo Research, Irvine, CA, USA) was used, and total RNA was treated with DNAse I (Invitrogen, Carlsbad, CA, USA) to eliminate any contaminating genomic DNA. Subsequently, first-strand cDNA was synthesized using the Sensifast cDNA kit (Bioline, London, NW2 6EW, UK) as per the manufacturer’s instructions, and stored at −20 °C prior to use.

### 2.3. Plasmid Preparation

Plasmid DNA for generating standard curves was prepared by carrying out serial 10-fold dilution steps and was used in both ddPCR and RT-qPCR assays. The target regions between positions 905 and 963 of the N protein-encoding gene were amplified, cloned into pGEM^®^-T Easy vector (Promega, Madison, WI, USA), and transformed into Competent TOP10 *Escherichia coli*, which were subsequently spread onto LB/ampicillin/IPTG/X-Gal plates and incubated overnight at 37 °C. The white colonies (*Escherichia coli* with inserted region) were selected for further growth in LB broth. The recombinant plasmid was extracted using the Plasmid Mini Kit (Zymo Research, Irvine, CA, USA) and stored at −20 °C prior to use. PCR and sequencing by Macrogen (Seoul, Republic of Korea) confirmed the inserted region’s presence and direction. The concentration of plasmid DNA was measured using a NanoDrop spectrophotometer (Thermo Fisher Scientific, Wilmington, DE, USA) and recalculated to plasmid copies/μL.

### 2.4. CDV Detection by Conventional Reverse-Transcription PCR

A 287 bp fragment of the nucleoprotein (NP), position 1055–1035 (Table 1), was amplified according to [28]. The amplification consisted of an initial denaturation step at 94 °C for 1 min, followed by 40 cycles consisting of a 1 min denaturation step at 94 °C, a 2 min annealing step at 59.5 °C, a 1 min extension step at 72 °C, and a final extension at 72 °C for 5 min, which was performed in a thermocycler (C1000 touch cycler; Bio-Rad, Hercules, CA, USA). Subsequently, the PCR product was loaded onto a 2% agarose gel stained with GoodView™ (SBS Genetech, Beijing, China). Electrophoresis was carried out at 100 V for a duration of 30 min. 

### 2.5. Real-Time Quantitative PCR

RT-qPCR was performed in a final volume of 20 μL. For each amplification reaction, 1 μL of cDNA was added to a reaction mixture containing 10 μL of oasig™ lyophilized 2× qPCR Mastermix (Genesig^®^, Chandler’s Ford Eastleigh, UK), 1.2 μL of primers/FAM labeled probe (600 nm/300 nm) (Table 1), and PCR-grade H_2_O up to the final volume (20 μL). The amplification process was performed using the Corbett Rotor-Gene 3000 (Corbett, Qiagen Rotor gene, Hilden, NW, Germany) according to the following thermal cycling conditions: 2 min at 95 °C for polymerase activation and DNA denaturation, 50 cycles of denaturation at 95 °C for 10 s, and annealing /extension + plate reading at 60 °C for 60 s. A non-template control was included in all runs, and every sample was measured in duplicate. Cq values and linear regression analyses of the calibration curves obtained with the tenfold serial dilutions with the standard plasmids were generated by Rotor-Gene AssayManager. For RT-qPCR, linear regression was applied to each of the three independent RT-qPCR experiments, and the correlation coefficient (R^2^), as well as PCR efficiency (E% = (10(−1/slope) − 1) × 100), were calculated. 

### 2.6. Droplet Digital PCR Assay 

DdPCR was performed according to the manufacturer’s instructions for the QX200 droplet digital PCR system (Bio-Rad, Hercules, CA, USA). The probe and primer sequences (Table 1) were designed to amplify a partial segment of the NP gene. The primer and probe concentrations and the annealing temperature and duration were optimized. Briefly, the 20 μL reaction mixture contained 10 μL master mix ddPCR for primers (600 nM) and probes (300 nM) and 1 μL cDNA. Initially, 20 μL of mix was transferred to a DG8 (Bio-Rad, Hercules, CA, USA) cartridge and 70 μL of Droplet generation oil was loaded into the DG8. Subsequently, a QX200 Droplet Generator (Bio-Rad, Hercules, CA, USA) was used to generate the partitions. A total of 40 μL of created droplets were loaded into a 96-well plate. The plate was run in a T100 Thermal Cycler (Bio-Rad, Hercules, CA, USA), with 40 cycles of amplification (20 s denaturation at 95 °C, followed by 30 s at 58 °C of annealing and extension). After amplification, droplets were individually read using a QX200 Droplet Reader (Bio-Rad, Hercules, CA, USA). The results were analyzed by QuantaSoft v.1.7.4 (Bio-Rad, Hercules, CA, USA). Reactions with a total number of droplets higher than 10,000 were considered. All samples were analyzed in triplicate, and a template-less control (NTC) reaction was also included. For the analysis of data obtained through ddPCR, we employed the Bio-Rad QX Manager™ Software v.1.7.4 and linear regression analysis. The threshold between positive and negative droplet populations was manually set, guided by per-plate positive controls and no-template controls.

### 2.7. Diagnostic Performance Evaluation and Data Analysis

Evaluation of analytical sensitivity: The analytical limit of detection (LoD) of RT-qPCR assays and ddPCR was evaluated through a series of 10-fold serial dilutions of the plasmid containing the NP gene. Each dilution was prepared in triplicate and subjected to testing by each assay on the same day. The LoD was determined as the highest dilution at which all replicates yielded positive, linear regression plots conducted with GraphPad Prism 7.00, and an analysis for LoD was conducted. Additionally, a linear regression analysis was performed to compare the RT-qPCR and ddPCR quantification with the theoretical dilutions of the plasmid.

Precision of RT-qPCR and ddPCR: Three dilutions of plasmid construction were prepared in triplicate and subjected to testing by RT-qPCR and ddPCR on the same day. The results are presented by the coefficients of variation (CV%). 

Diagnostic Performance: The sensitivity and specificity of the molecular techniques (PCR conventional, RT-qPCR, and ddPCR) were determined using the two-by-two table [30,31] against the clinical diagnosis of the 92 samples. Kappa indices were calculated using WinEpi Software (Working in Epidemiology, http://www.winepi.net/. Receiver operating characteristic (ROC) curves were constructed to evaluate the diagnostic performance of the RT-PCR, RT-qPCR, and ddPCR assays. RStudio, with the pROC package, was employed to calculate the area under the curve (AUC) along with the corresponding 95% confidence interval (95% CI). To assess the significance of differences in CDV-positive case detection between PCR techniques, chi-square tests were performed. The proportions of positive cases detected by ddPCR were compared against conventional PCR and RT-qPCR separately. All statistical analyses were conducted using GraphPad Prism 7.00. The significance level was set at *p* < 0.05. For each comparison, chi-square values and *p*-values were calculated to determine the statistical significance of the observed differences in detection rates.

## 3. Results

### 3.1. Description of Clinical Signs in Canines Diagnosed with Canine Distemper 

Of the 76 samples collected from canines diagnosed with canine distemper, 16% were from dogs aged 0–1 year, 22% were from dogs 1–7 years old, and 8% were from dogs older than 7 years, with 9% lacking age data. The sex distribution was 43% male and 45% female, with 12% unrecorded. Nine breeds were represented, with mixed-breed dogs being most common (42.6%), followed by Poodles (11%), Labradors (7%), French Bulldogs (7%), Pugs (4.2%), German Shepherds (4.2%), Pitbulls (4.2%), Beagles (3%), and Shar-Peis (3%). Additionally, 18% of the cases lacked breed information. Regarding vaccination status, 21% were up to date, 49% were unvaccinated, and 30% had unknown status. Figure 1 illustrates the frequency of the primary clinical signs observed, highlighting the predominance of nervous and digestive symptoms in dogs diagnosed with canine distemper. Additionally, 22 brain tissue samples were analyzed, with viral presence confirmed through immunohistochemistry. Histological examination of the tissue sections, stained with H&E and LFB, was performed to assess the damage. We observed that 27% of the cases showed acute lesions, characterized by focal vacuolation, minimal gliosis, minimal perivascular cuffing, the presence of inclusion bodies, and CDV-positive cells. Subacute lesions were observed in 55% of the cases and were characterized by patchy demyelination, extensive gliosis, neuronal necrosis, prominent perivascular cuffing, prominent CDV inclusion bodies, and numerous CDV-positive cells. Chronic lesions represented 18% of the samples included in this study, characterized by lesions like those of the subacute stage but with increased neuronal degeneration (Appendix A).

### 3.2. Analytical Sensitivity of RT-qPCR and ddPCR

Serial 10-fold dilutions of the plasmid from 4.22 × 10^12^ copies/μL to 1 × 10^−4^ copies/μL were used to determine the detection limit of CDV in RT-qPCR and ddPCR. Quantification threshold (Ct) values were measured in triplicate and were plotted against the known copy numbers of plasmid dilution. The generated standard curve showed good linearity (slope = −3.366), with an efficiency (E) of 115% and a coefficient of linear regression R^2^ = 0.9235. The limit of detection in RT-qPCR was 86 copies/μL.

To assess the detection limit in ddPCR, we employed the same serial dilutions of the plasmid as those in RT-qPCR. Positive droplets were determined by fluorescence intensity using QuantaSoft software v1.7.4. The software provides automated analysis and manual thresholding tools, facilitating the accurate identification of positive and negative droplet populations. Positive droplets were defined based on their fluorescence intensity with only droplets that exceeded the threshold value being counted as positive. The analytical curve of the proposed method for quantification of CDV by ddPCR was linear over the entire tested range. Through linear regression analysis (R^2^ = 0.953), we calculated a sensitivity (slope) of 0.6417 and an intercept of −0.5154 (Figure 2). At a 95% confidence level, we calculated a limit of detection (LoD) of 3 copies/µL of reaction volume and a limit of quantification (LoQ) of 8 copies/µL of reaction volume. For the qualitative analysis, samples with a ddPCR result of ≥3 copies/µL of reaction were classified as positive.

The linear regression analysis comparing plasmid quantification between RT-qPCR and ddPCR against the theoretical dilutions revealed differences in slopes and intercepts. For RT-qPCR, the regression equation was Y = 1.106*X + 0.1737, indicating a slope close to 1, which strongly aligns with the theoretical dilutions. The slight positive intercept of 0.1737 may indicate a minor overestimation in the more diluted samples. In contrast, the ddPCR regression equation was Y = 0.6417*X − 0.5154. Although the slope is less than 1, indicating a slight tendency to underestimate quantification compared to theoretical dilutions, the negative intercept of −0.5154 highlights ddPCR’s ability to accurately quantify even at very low concentrations. The coefficient of determination (R^2^) values were close to 1 for both methods: 0.947 and 0.953 for RT-qPCR and ddPCR, respectively, further supporting the high accuracy of these methods (Figure 3).

### 3.3. Precision of RT-qPCR and ddPCR

The precision of RT-qPCR and droplet digital PCR was evaluated using three independent dilutions of a plasmid construction. The coefficient of variation (CV) and standard deviation (SD) were calculated to assess the reproducibility of each method.

The CV values across the dilutions were significantly higher for RT-qPCR, indicating greater variability among technical replicates. Specifically, the CV ranged from 98% to 150%. These results underscore the inherent variability in RT-qPCR, which can impact the accuracy of viral load quantification, especially at low concentrations. In contrast, ddPCR demonstrated markedly improved precision. The CV values for ddPCR were consistently lower, ranging from 8.9% to 77.9% (Table 2). 

### 3.4. Comparing the Performance of Conventional RT-PCR, RT-qPCR, and ddPCR Against the Clinical Diagnosis 

In this study, a total of 92 samples were used. Seventy-six of these samples came from dogs clinically diagnosed with CDV by experienced veterinarians, while sixteen samples were from healthy dogs (negative controls). Each sample was tested in duplicate. Our research, which focused on samples with a clinical diagnosis of CDV disease, compared the performance of conventional RT-PCR, RT-qPCR, and ddPCR. Conventional RT-PCR detected 42.4% of samples as CDV-positive and 57.61% as negative, including the sixteen non-symptomatic CDV (control) samples. In contrast, RT-qPCR identified 47.9% as positive and 52.2% as negative, including the control samples. When comparing the clinical diagnosis with ddPCR, we observed that 59.8% of the samples were detected as positive and 40.2% as negative, including the control samples (Table 3 and Table 4).

The ROC curve analysis, a reliable method, revealed the following areas under the curve (AUCs) for the different PCR techniques compared to the clinical diagnosis of canine distemper: conventional PCR (RT-PCR) showed an AUC of 0.76, indicating moderate diagnostic accuracy. RT-qPCR exhibited a higher AUC of 0.88, reflecting a good level of diagnostic performance. Droplet digital PCR demonstrated an AUC of 0.87, indicating good accuracy and performance comparable to real-time PCR (RT-qPCR) in diagnosing canine distemper based on the clinical criteria (Figure 4).

The proportions of positive cases detected by ddPCR compared to those found by conventional RT-PCR were significantly higher (*p* = 0.01234), and when we compare with RT-qPCR, we see that it detects more in ddPCR with a significant trend (*p* = 0.09529). Digital PCR detected the highest number of CDV-positives in brain samples, with all 22 samples (100%) testing positive, most with high viral loads (Figure 5A). Additionally, 60% of blood samples were identified as positive by ddPCR, a higher percentage compared to 33.3% detected by conventional RT-PCR and 50% by RT-qPCR. We identified CDV at low viral loads using ddPCR in eleven blood samples, one nasal secretion, and three nervous tissue samples (Figure 5B). In contrast, the other PCR methods did not detect the virus in these samples.

## 4. Discussion

CDV continues to threaten carnivorans, particularly domestic dogs, emphasizing the critical necessity for accurate and prompt diagnostic methodologies to ensure effective management. The definitive identification of CDV relies primarily on viral detection through PCR analysis of whole blood, serum, and cerebrospinal fluid [28]. The variability in the viral load excreted by affected animals, contingent upon the progression and stage of the infection, often presents a diagnostic challenge. The discernment of the virus may be hindered by the limited presence of viral material or the interference of inhibitors within the sampled specimens. Against this backdrop, our study explored the diagnostic prowess of ddPCR, contrasting its efficacy with conventional RT-PCR and quantitative real-time RT-PCR. By employing clinical samples from symptomatic dogs, this investigation aimed to shed light on the potential advancements ddPCR offers in overcoming the inherent challenges associated with accurate and timely CDV diagnosis.

In terms of analytical performance, the determination of the LoD for CDV using a plasmid offers crucial insights into the sensitivity of the diagnostic methods, specifically RT-qPCR and ddPCR. Our study established a detection limit of 86 copies/μL for RT-qPCR, which aligns with previous findings in the literature. Elia et al. [17] reported a detection limit of 10^2^ genomic copies/µL, while Sehata et al. [32] and Brown et al. [33] calculated an LoD of 50 copies/µL using TaqMan probes targeting the phosphoprotein gene. Halecker et al. [34] reported a 10^1^ genomic copies/µL detection limit, and Geiselhardt et al. [35] calculated an LoD of 5.47 copies ± 2.49 copies. Although the interlaboratory variability of quantitative viral PCR results in virus diagnosis has not undergone comprehensive analysis, we expected to observe 30 to 40% or more variance when evaluating RT-qPCR results for viral targets. This variability may be influenced by multiple factors, including the choice of quantitative calibrator, detection reagents, nucleic acid extraction methods, and the molecular amplification target gene, among other variables [36]. In contrast, ddPCR offers a significant advantage by eliminating the need for standard curves, which can introduce variability and require additional preparation. This simplification of the workflow reduces potential sources of error, enhancing the reliability and efficiency of the analysis [37].

In this study, we developed a novel method of detecting CDV employing ddPCR. This method enables absolute quantification of the virus across various sample types (urine, blood, nasal and eye swabs, and cerebellum) with high sensitivity (72.4%) and specificity (100%). Compared to RT-qPCR, our results revealed a lower detection limit in ddPCR at 3 copies/μL and an area under the curve of 0.87. The linear regression equation demonstrated a robust correlation between the observed quantification and the theoretical dilutions, highlighting the method’s consistency across a range of concentrations. The slope of 0.6417 suggests a slight underestimation compared to the expected values, which could be attributed to the inherent precision of ddPCR at lower concentrations. These results highlight ddPCR’s sensitivity and effectiveness in quantifying low-abundance targets. This high sensitivity, as revealed by the comparative data from previous studies, has significant implications for virology. For example, Souto et al. [38] determined the limit of detection and quantification to be 10 copies per reaction for viral nervous necrosis virus (VNNV), an RNA virus belonging to the family *Nodaviridae*. Consistent with our results, Hui et al. [39] established the limit of detection for the RNA Zika virus at 1 copy/μL. 

The ddPCR highlighted a significant reduction in the coefficient of variation (CV) when compared to that with RT-qPCR, with values ranging from 8.9% to 77.9%. This marked improvement demonstrates ddPCR’s superior reproducibility, particularly in detecting low viral loads. In the context of diagnosing CDV, this reduced variability is crucial, as it suggests that ddPCR can provide more consistent and reliable results, especially in cases where the viral load is near the detection threshold. The enhanced precision of ddPCR is particularly advantageous for quantifying low-abundance targets, where even minor variations can lead to significant discrepancies in viral load estimation. Multiple researchers have consistently documented this significant finding. For example, Ciesielski et al. [40] compared the sensitivity, quantification limits, and reproducibility of two similar workflows, qPCR and ddPCR, by analyzing 60 raw wastewater samples from nine treatment plants over six months. Both methods detected increasing SARS-CoV-2 concentrations, with ddPCR showing greater analytical sensitivity and a lower limit of detection (0.066 copies/μL) compared to qPCR (12 copies/μL). Another study explored the use of ddPCR for quantifying HIV DNA in patients on antiretroviral therapy [21]. Their results demonstrate ddPCR’s superior precision and accuracy compared to real-time PCR, showing a five-fold reduction in variability for total HIV DNA. It is important to note that various research groups have been utilizing ddPCR for over a decade to quantify and detect viruses, consistently yielding promising results [41]. The application of ddPCR for CDV detection aligns with its successful utilization in detecting various other viruses, as evidenced by studies on influenza virus [42], hepatitis B virus [43], SARS-CoV-2 [44,45], foot-and-mouth disease virus [46], human parechoviruses (HPeVs) [47], bovine leukemia virus (BLV) [48], and cytomegalovirus [49]. This broader applicability underscores the versatility and reliability of ddPCR across different viral pathogens. 

It is important to acknowledge that our study, like many field studies in veterinary virology, faces the challenge of not having a universally accepted gold standard for CDV RNA detection. We relied on clinical diagnosis by experienced veterinarians as our reference point, which reflects real-world scenarios but may have limitations. While RT-qPCR is generally considered the most sensitive and specific method available in clinical practice, definitive confirmation would ideally involve a combination of methods, including RT-PCR followed by sequencing, virus isolation in cell culture, and immunohistochemistry on tissue samples. However, such comprehensive testing is often impractical in large-scale clinical studies due to time and resource constraints. We recognize this as a potential limitation of our study. Future research could benefit from more comprehensive confirmatory testing on a subset of samples to further validate the performance of ddPCR in CDV detection. Despite this limitation, we believe our approach provides valuable insights into the relative performance of different PCR methods in a clinical context, offering a realistic assessment of their utility in veterinary practice. This is particularly relevant given that in this study, the use of ddPCR led to a 25% increase in the detection of positive cases for CDV compared to conventional diagnostic methods. This heightened sensitivity carries significant clinical, diagnostic, and epidemiological implications, even if it cannot provide absolute confirmation of CDV RNA presence. We believe this approach still provides valuable insights into the relative performance of different PCR methods in a clinical context, while addressing the limitations of our study and highlighting the practical implications of our findings.

In this study, the use of ddPCR led to a statistically significant increase of 41% in the detection of positive cases for CDV compared to conventional PCR. Additionally, a 25% increase in detection was observed compared to that with RT-qPCR, although this difference was not statistically significant at the conventional threshold. This heightened sensitivity carries significant clinical, diagnostic, and epidemiological implications. Clinically, the improved diagnosis provided by ddPCR gives us confidence in our ability to identify infected animals, reducing the risk of misdiagnosis and ensuring timely treatment. In shelters, misdiagnosis could result in admitting significantly more animals with active infections, posing a severe risk to unvaccinated populations and potentially causing widespread outbreaks. From a diagnostic standpoint, ddPCR’s higher sensitivity could be a game-changer in settings where cost efficiency is critical, such as in shelters, by allowing pooled sample testing without compromising accuracy. This approach could significantly reduce the cost per test while maintaining high detection rates. Moreover, the technique’s application in environmental sampling, such as in parks and shelters, could provide valuable epidemiological insights, particularly due to its high sensitivity and ability to detect low viral loads. This could enable more effective monitoring of CDV’s presence in the environment, helping to prevent outbreaks before they occur.

## 5. Conclusions

Our study demonstrates that ddPCR offers a unique advantage in the early and precise detection of CDV, particularly in cases where the viral load may be low. This heightened sensitivity opens new possibilities for the effective management of CDV, including improved surveillance, earlier intervention, and more targeted control strategies. Furthermore, the high sensitivity of ddPCR could be particularly valuable for monitoring CDV in wildlife populations, where sample collection is often challenging, and viral loads may be low. These findings emphasize the practical implications of our research for veterinary medicine, animal welfare efforts, and wildlife conservation, potentially contributing to a more comprehensive approach to CDV management across both domestic and wild animal populations.

## Figures and Tables

**Figure 1 viruses-16-01720-f001:**
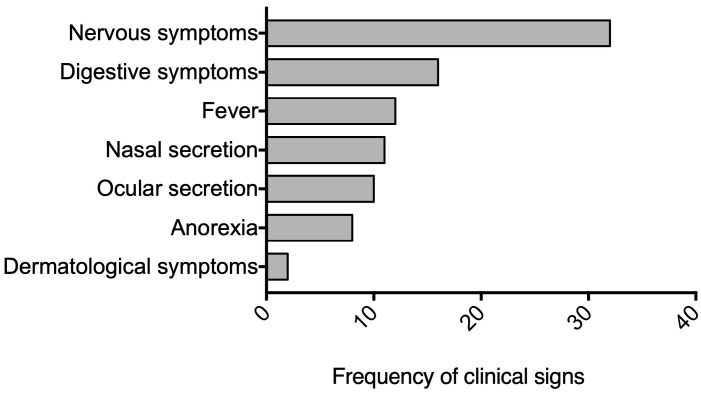
Frequency of clinical signs observed in 76 dogs diagnosed with canine distemper virus (CDV). Nervous symptoms were the most prevalent, followed closely by digestive symptoms. Nervous symptoms included seizures, muscle twitches, partial or total paralysis, and abnormal behaviors. Digestive symptoms encompassed vomiting, diarrhea leading to dehydration, loss of appetite resulting in weight loss, lethargy, and depression. Dermatologic symptoms, primarily characterized by hyperkeratosis of the paw pads and nose, were also observed. This distribution of symptoms highlights the multi-systemic nature of CDV infection, with a particular emphasis on neurological and gastrointestinal manifestations.

**Figure 2 viruses-16-01720-f002:**
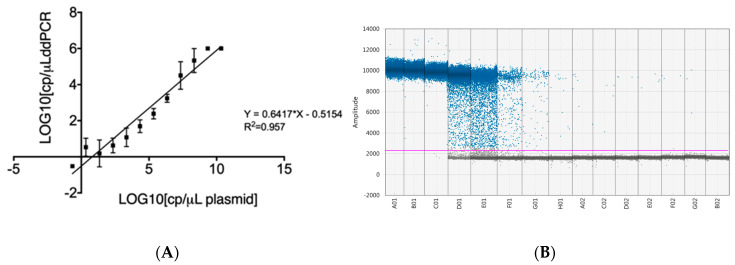
Limit of detection of ddPCR. (**A**) Log10 copies/μL serial dilutions of the plasmid containing the CDV nucleoprotein (NP) gene. (**B**) Scatter diagram from droplet digital PCR of plasmid dilutions ranging from 2.18 × 10^10^ to 2.18 × 10^−4^ (A01–G02) and no-template control (NTC) (B02). Gray dots represent negative events; blue dots represent positive events. The purple line indicates the threshold for positive signals.

**Figure 3 viruses-16-01720-f003:**
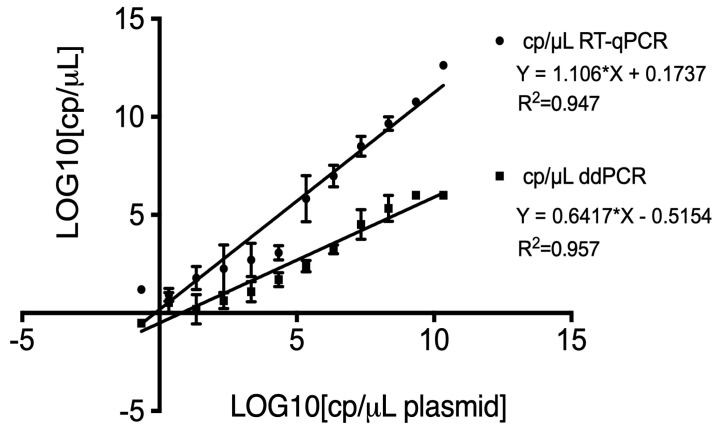
Linear regression analysis of plasmid quantification using RT-qPCR (point) and ddPCR (square) compared to theoretical dilutions. The equations for the regression lines were Y = 1.106X + 0.1737 (R^2^ = 0.947) for RT-qPCR and Y = 0.6417X − 0.5154 (R^2^ = 0.953) for ddPCR. The x-axis represents the log10 of theoretical copy numbers, while the y-axis represents the log10 of measured copy numbers.

**Figure 4 viruses-16-01720-f004:**
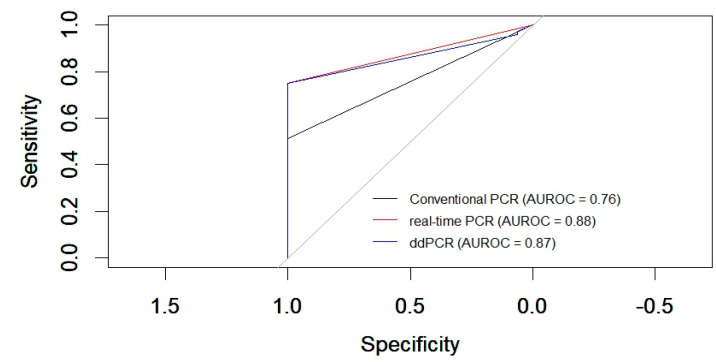
Comparison of receiver operating characteristic (ROC) curves of conventional PCR, real-time PCR, and ddPCR for CDV. The black curve represents conventional PCR, the red curve corresponds to RT-qPCR, and the blue curve represents digital PCR.

**Figure 5 viruses-16-01720-f005:**
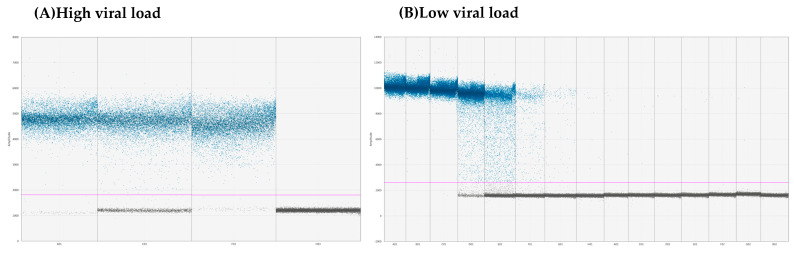
Scatter diagram from droplet digital PCR: (**A**) ddPCR results for samples with high viral loads (5127, 2154, and 5189 copies/μL, respectively). The droplet distribution shows an increase in positive droplets (blue dots) compared to negative ones (gray dots). The high concentration of blue dots reflects a high viral load, with a larger proportion of droplets exceeding the fluorescence threshold. H03 represents the no-template control (NTC). (**B**) ddPCR results for samples with low viral loads (46.6, 6, 40.7, 20.2, 5.1, 18.7, 4.62, 24.6, 7.6, and 3.95 copies/μL, respectively). The droplet distribution is shown with a high number of negative droplets (gray dots) and a small number of positive droplets (blue), indicating the presence of the virus at low concentrations. The purple line indicates the threshold for positive signals.

**Table 1 viruses-16-01720-t001:** Primers and probe used in this study.

Application	Primer	Sequence (5′ to 3′)
RT-PCR [28]	CDV-NP_reverse	CAAGATAACCATGTACGGTGC
	CDV-NP_forward	ACAGGATTGCTGAGGACCTAT
RT-qPCR and ddPCR [29]	CDV-reverse	ATGAGTTTTCCGGAGAATTAACAA
	CDV-forward	AGCTAGTTTCATCCTAACTATCAAGT
	CDV probe	FAM-TGGCATTGAAACTATGTATCCGGCTCT-BHQ1-3

**Table 2 viruses-16-01720-t002:** Precision of RT-qPCR and ddPCR of three independent dilutions of the plasmid construction. SD (standard deviation); CV (%) (coefficient of variation).

	RT-qPCR Assay			ddPCR Assay		
Plasmid dilution ^1^	10^−4^	10^−6^	10^−8^	10^−4^	10^−6^	10^−8^
Mean of plasmid copy number ^2^	6.9 × 10^6^	1.1 × 10^7^	1.6 × 10^2^	2.8 × 10^3^	5.7 × 10^2^	2.0 × 10^1^
SD	2990	864.3	28.9	84.8	50.8	59.9
CV (%)	150.3	148.8	99	8.9	6.9	77.9

^1^ Three independent dilutions. ^2^ Mean of the assay performed in triplicate.

**Table 3 viruses-16-01720-t003:** Comparison of the PCR conventional, RT-qPCR, and ddPCR results across 92 different sample types. n = total number of samples.

	Blood * (n = 57)	Urine * (n = 9)	Nasal and Eye Swabs * (n = 4)	Brain * (n = 22)	Total Positive	Total Negative
RT-PCR	14	3	3	19	39	53
RT-qPCR	21	4	3	16	44	48
ddPCR	24	5	4	22	55	37

* Number of positive samples.

**Table 4 viruses-16-01720-t004:** Comparison between clinical diagnostics with RT-PCR, RT-qPCR, and ddPCR results of 92 samples (TPV: true-positive value; TNV: true-negative value; FNV: false-negative value; FP: false positive, FN: false negative).

	True Result (TPV + TNV)	False Result (FPV + FNV)	FP (%)	FN (%)	Sensitivity (%)	Specificity (%)	Kappa (IC 99.5%)
RT-PCR	55	37	0	48.7	51.3	100	0.268
RT-qPCR	60	32	0	42.1	57.9	100	0.324
ddPCR	71	21	0	27.6	72.4	100	0.477

## Data Availability

The data presented in this study are available on request from the corresponding author.

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
