# Peer review of "Droplet Digital PCR Enhances Sensitivity of Canine Distemper Virus Detection"

_viruses, 2024, doi:10.3390/v16111720_

Round 1
Reviewer 1 Report
Comments and Suggestions for Authors
Title – rewrite to read as: Droplet digital PCR enhances sensitivity canine distemper virus detection
Line 13 – word “carnivore” refers to consumption of animal tissues; “carnivoran” indicates order Carnivora and would better fit – change accordingly throughout the manuscript
Line 15 – replace efficacy with performance
Line 18 – replace symptoms with clinical signs
Line 19 – replace asymptomatic with “apparently healthy”
Use only one decimal place for percentages, e.g. 42.4% (instead of 42.39%)
Line 27 – replace concordance with “agreement”
Keywords – display alphabetically
Lines 32-34 – rewrite to read as: Canine distemper virus (CDV), belongs to the genus Morbillivirus within the family Paramyxoviridae and is an enveloped negative polarity RNA virus [1].
Display genus and family names in italics
Line 37 – replace Cetacean with cetacean
Line 39 – adapt to red as: CDV infects a wide range of CARNIVORANS, including members of THE FAMILIES Canidae, Procyonidae…
Is there any reason for displaying families in this order? Otherwise, display them alphabetically
Line 41 – replace [2, 3, 4, 5] with [2–5]
Line 41 – delete “rate” – fatality is not a true rate
Line 42 – adapt: The disease caused by CDV is multi-systemic, affecting the respiratory, …
Line 43 – clinical signs (instead of signs only)
Line 45 – correct the presentation of reference [8] – change accordingly throughout the manuscript (lines 60, 65, etc.)
Line 50 – neurological SIGNS (delete symptoms)
Line 51 – delete “disease”
Line 55 – Rapid and ACCURATE diagnosis of CDV DISEASE facilitates…
Line 57 – replace reliable with “valid”
Lines 57-59 – revise: Variable signs of distemper and subclinical status of the INFECTION in some cases are the main challenges of DIAGNOSIS based on PHYSICAL examination.
Lines 91-92 – why these numbers of samples, i.e. 76 and 16? Which criteria have been followed to validate these numbers?
Are the authors sure that those 76 dogs had CDV disease? Were they diagnosed or were they clinically suspect?
Line 92, etc. – CDV is not a disease – the disease is distemper or CDV disease – please change accordingly throughout the manuscript
Line 95 – replace subclinical disease with subclinical infection
Line 104 – replace symptoms with clinical signs – change accordingly throughout the manuscript
Line 105 – adapt: coughing that COULD be accompanied…
Line 110 – replace symptomatology with clinical manifestations
Line 112 – display reference in ascending order
Line 115 – avoid using “we”
Line 116 – identification of companies should include city and country (an abbreviation of state may also be included) – change accordingly throughout the manuscript
Line 210 – replace Clinical Performance with Diagnostic Performance
Lines 220-229 – these data (including Figure 1) may be regarded as belonging to the materials and methods section as they describe dogs that were seen by specialized veterinarians before their testing with RT-PCR, RT-qPCR and ddPCR for CDV detection
Line 249 – 3.2. Analytical sensitivity of RT-qPCR and ddPCR – please do not repeat information from the materials and methods section
Table 4 – one decimal place for percentages is enough
Line 352 – carnivorans
Line 353 – replace precise with accurate
Discussion – please inform on whether differences (especially percentages) are statistically significant or not.
For example, lines 420-421 – is this difference of 17% statistically significant?
References – please, limit the number of authors; use lowercase initials in titles, whenever possible; adapt reference 3.
Author Response
|
Comments 1: [Title – rewrite to read as: Droplet digital PCR enhances sensitivity canine distemper virus detection] |
|
Response 1: Thank you for pointing this out. We agree with this comment. Therefore, we have modified the title in page 1, line 2. |
|
Comments 2: [Line 13 – word “carnivore” refers to consumption of animal tissues; “carnivoran” indicates order Carnivora and would better fit – change accordingly throughout the manuscript] |
|
Response 2: Thank you for pointing this out. We agree with this comment. We have, accordingly, modified the word carnivore to carnivoran to emphasize this point. This change can be found on page, page 1, line 13 and line 40; page 10, line 353.
Comments 3: Line 15 – replace efficacy with performance Response 3: Thank you for pointing this out. We agree with this comment. Therefore, we have modified page 1, line 15.
Comments 4: Line 19 – replace asymptomatic with “apparently healthy” Response 4: Thank you for pointing this out. We agree with this comment. Therefore, we have modified page 1, line 19.
Comments 5: Use only one decimal place for percentages, e.g. 42.4% (instead of 42.39%) Response 5: Agree. We have, accordingly, changed to emphasize this point. Therefore, we have modified the page 1, line 26; page 8, line 301; table 2 on page 8; page 8, lines 312, 314 and 316; table 4 on page 9; page 10, line 384; page 11 line 398.
Comments 6: Line 27 – replace concordance with “agreement” Response 6: Thank you for pointing this out. We agree with this comment. Therefore, we have modified page 1, line 27.
Comments 7: Keywords – display alphabetically Response 7: Thank you for pointing this out. We agree with this comment. Therefore, we have modified page 1, line 30.
Comments 8: Lines 32-34 – rewrite to read as: Canine distemper virus (CDV), belongs to the genus Morbillivirus within the family Paramyxoviridae and is an enveloped negative polarity RNA virus [1]. Response 8: Agree. We have, accordingly, changed to emphasize this point. Therefore, we have modified page 1, line 32-34.
Comments 9: Display genus and family names in italics Response 9: Thank you for pointing this out. We agree with this comment. Therefore, we have modified the page 1, line 33-34, line 40-42; page 11, line 395
Comments 10: Line 37 – replace Cetacean with cetacean Response 10: Thank you for pointing this out. We agree with this comment. Therefore, we have modified page 1, line 37.
Comments 11: Line 39 – adapt to red as: CDV infects a wide range of CARNIVORANS, including members of THE FAMILIES Canidae, Procyonidae Is there any reason for displaying families in this order? Otherwise, display them alphabetically Response 11: Thank you for pointing this out. We agree with this comment. Therefore, we have modified page 1, line 39-40.
Comments 12: Line 41 – replace [2, 3, 4, 5] with [2–5] Response 12: Thank you for pointing this out. We agree with this comment. Therefore, we have modified page 1, line 41.
Comments 13: Line 41 – delete “rate” – fatality is not a true rate Response 13: Thank you for pointing this out. We agree with this comment. Therefore, we have modified page 1, line 42.
Comments 14: Line 42 – adapt: The disease caused by CDV is multi-systemic, affecting the respiratory, Response 14: Thank you for pointing this out. We agree with this comment. Therefore, we have modified page 1, line 43-44.
Comments 15: Line 43 – clinical signs (instead of signs only) Response 15: Thank you for pointing this out. We agree with this comment. Therefore, we have modified page 1, line 45.
Comments 16: Line 45 – correct the presentation of reference [8] – change accordingly throughout the manuscript (lines 60, 65, etc.) Response 16: Thank you for pointing this out. We agree with this comment. Therefore, we have modified page 2, line 47, line 61 and line 66.
Comments 17: Line 50 – neurological SIGNS (delete symptoms) Response 17: Thank you for pointing this out. We agree with this comment. Therefore, we have modified page 2, line 51.
Comments 18: Line 51 – delete “disease” Response 18: Thank you for pointing this out. We agree with this comment. Therefore, we have modified page 2, line 52.
Comments 19: Line 55 – Rapid and ACCURATE diagnosis of CDV DISEASE facilitates… Response 19: Thank you for pointing this out. We agree with this comment. Therefore, we have modified page 2, line 56.
Comments 20: Line 57 – replace reliable with “valid” Response 20: Thank you for pointing this out. We agree with this comment. Therefore, we have modified page 2, line 57.
Comments 20: Line 57 – replace reliable with “valid” Response 20: Thank you for pointing this out. We agree with this comment. Therefore, we have modified page 2, line 58.
Comments 21: Lines 57-59 – revise: Variable signs of distemper and subclinical status of the INFECTION in some cases are the main challenges of DIAGNOSIS based on PHYSICAL examination. Response 21: Thank you for pointing this out. We agree with this comment. Therefore, we have modified page 2, line 59-60.
Comments 22: Lines 91-92 – why these numbers of samples, i.e. 76 and 16? Which criteria have been followed to validate these numbers? Are the authors sure that those 76 dogs had CDV disease? Were they diagnosed or were they clinically suspect? Response 22: We appreciate the reviewer's questions regarding our sample selection and CDV diagnosis. The number of samples was not predetermined but rather accumulated based on availability during the study period. Our sample size is comparable to or exceeds those used in similar studies evaluating diagnostic methods for CDV. For instance, [Ciesielski et al., 2021; Aizawa et al., 2016]. While not based on a formal power calculation, we believe this sample size provides meaningful results for comparing the diagnostic techniques.
Regarding the CDV status, the 76 dogs included in the study were clinically diagnosed with presumptive CDV by experienced veterinary clinicians based on characteristic symptoms and clinical presentation. It's important to note that definitive ante-mortem diagnosis of CDV is challenging, as clinical signs can vary and may resemble other canine respiratory and neurological diseases. Our study aimed to compare molecular diagnostic techniques against this clinical diagnosis, which represents a real-world scenario faced by veterinarians.
The dogs exhibited multiple clinical signs consistent with CDV, such as respiratory signs (nasal discharge), ocular discharge, gastrointestinal symptoms, and neurological manifestations. Additionally, most of these dogs had an unknown or incomplete vaccination status, which is a known risk factor for CDV infection. While we acknowledge that laboratory confirmation is ideal, our study's objective was to evaluate these molecular techniques as potential improvements over clinical diagnosis alone. We believe this approach provides valuable insights into the performance of these molecular techniques in a clinically relevant context, where veterinarians must often make treatment decisions based on presumptive diagnosis.
Comments 23: Line 92, etc. – CDV is not a disease – the disease is distemper or CDV disease – please change accordingly throughout the manuscript Response 23: Thank you for pointing this out. We agree with this comment. Therefore, we have modified page 2, line 93,94; page 8, line 312.
Comments 24: Line 95 – replace subclinical disease with subclinical infection Response 24: Thank you for pointing this out. We agree with this comment. Therefore, we have modified page 2, line 96.
Comments 25: Line 104 – replace symptoms with clinical signs – change accordingly throughout the manuscript Response 25: Thank you for pointing this out. We agree with this comment. Therefore, we have modified the page 3, line 106, 111; page 5, line 221
Comments 26: Line 105 – adapt: coughing that COULD be accompanied… Response 26: Thank you for pointing this out. We agree with this comment. Therefore, we have modified page 3, line 107.
Comments 27: Line 110 – replace symptomatology with clinical manifestations Response 27: Thank you for pointing this out. We agree with this comment. Therefore, we have modified page 3, line 112.
Comments 28: Line 112 – display reference in ascending order Response 28: Thank you for pointing this out. We agree with this comment. Therefore, we have modified the page 3, line 114 Comments 29: Line 115 – avoid using “we” Response 29: Thank you for pointing this out. We agree with this comment. Therefore, we have modified page 3, line 117. Comments 30: Line 116 – identification of companies should include city and country (an abbreviation of state may also be included) – change accordingly throughout the manuscript Response 30: Thank you for pointing this out. We agree with this comment. Therefore, we have modified the page 3; line 118, 119, line 137-139, line 141, line 147; page 4; line 151, 152, 163, 171, 172 Comments 31: Line 210 – replace Clinical Performance with Diagnostic Performance Response 31: Thank you for pointing this out. We agree with this comment. Therefore, we have modified page 5, line 212.
Comments 32: Lines 220-229 – these data (including Figure 1) may be regarded as belonging to the materials and methods section as they describe dogs that were seen by specialized veterinarians before their testing with RT-PCR, RT-qPCR and ddPCR for CDV detection Response 32: We appreciate the reviewer's suggestion and understand the rationale behind it. However, we believe the data presented in lines 220-229 and Figure 1 are more appropriately placed in the Results section. This information represents findings from our analysis of the clinical records rather than part of the study design or methodology. These results provide important insights into the clinical presentation of the dogs in our study, which is crucial for understanding the context of our molecular diagnostic comparisons. The frequency and distribution of clinical signs observed in the 76 dogs diagnosed with presumptive CDV are outcomes of our investigation, not pre-established criteria for sample selection. Furthermore, this information serves to characterize the study population in detail and demonstrate the multi-systemic nature of CDV infection in our cohort. It also provides context for interpreting the performance of the molecular diagnostic techniques we evaluated. Presenting this data in the Results section allows for a more logical flow of information, from clinical observations to molecular diagnostic findings.
Comments 33: Line 249 – 3.2. Analytical sensitivity of RT-qPCR and ddPCR – please do not repeat information from the materials and methods section Response 33: We appreciate the reviewer's concern about potential repetition of information. However, after careful review, we respectfully disagree that section 3.2 repeats information from the Materials and Methods section. While the Materials and Methods section describes the procedures and theoretical aspects of the plasmid preparation and assay setup, section 3.2 presents the actual results and analysis of these experiments, including: The specific detection limits found for both RT-qPCR and ddPCR. The detailed performance characteristics of each assay (e.g., linearity, efficiency, R² values). The comparative analysis of plasmid quantification between RT-qPCR and ddPCR against theoretical dilutions. These results are crucial for understanding the analytical sensitivity of both methods and provide essential context for interpreting the subsequent findings in our study. We believe this information is appropriately placed in the Results section and does not constitute a repetition of the methodological details.
Comments 34: Table 4 – one decimal place for percentages is enough Response 34: Thank you for pointing this out. We agree with this comment. Therefore, we have modified table 4.
Comments 35: Line 352 – carnivorans Response 35: Thank you for pointing this out. We agree with this comment. Therefore, we have modified the line 354.
Comments 36: Line 353 – replace precise with accurate Response 36: Thank you for pointing this out. We agree with this comment. Therefore, we have modified the line 355.
Comments 37: Discussion – please inform on whether differences (especially percentages) are statistically significant or not. For example, lines 420-421 – is this difference of 17% statistically significant? Response 37: Thank you for this important observation. We appreciate your attention to detail regarding the statistical significance of our findings. In response to your comment, we have conducted a thorough statistical analysis of our data using chi-square tests to compare the detection rates between ddPCR and conventional methods (conventional PCR and RT-qPCR). Our analysis revealed that: The increase in detection rate of ddPCR compared to conventional PCR (41% increase) is statistically significant (χ² = 6.2609, p = 0.01234). The increase in detection rate of ddPCR compared to RT-qPCR (25% increase) approaches but does not reach conventional statistical significance (χ² = 2.7826, p = 0.09529). We have revised the discussion section to reflect these findings, providing a more accurate representation of our results. Specifically, we have modified lines 454-470. We have also added a brief description of the statistical analysis in the Materials and Methods section to ensure transparency and reproducibility of our findings.
Comments 38: References – please, limit the number of authors; use lowercase initials in titles, whenever possible; adapt reference 3. Response 38: Thank you for pointing this out. We agree with this comment. Therefore, we have modified the number or authors in reference.
|
|
4. Response to Comments on the Quality of English Language |
|
Point 1: |
|
Response 1:
|
|
5. Additional clarifications |
|
We would like to express our sincere gratitude to the reviewers for their thorough and insightful comments. Their expertise and attention to detail have significantly enhanced the quality and clarity of our manuscript. Following the reviewers' suggestions, we have made comprehensive revisions throughout the paper. These changes have improved the precision of our terminology, the clarity of our methodology, and the presentation of our results. We believe that these revisions have strengthened our manuscript, making it more accessible and valuable to the readership of VIRUSES. The process of addressing the reviewers' comments has also led us to refine our analysis and sharpen our conclusions. We are confident that the revised manuscript now presents a more robust and nuanced examination of droplet digital PCR in canine distemper virus detection. We appreciate the opportunity to contribute to the field of virology through VIRUSES. We believe that our study, with these revisions, offers important insights into molecular diagnostic techniques for CDV, which may have broader implications for viral detection methodologies. Once again, we thank the reviewers and editors for their time and expertise. We are grateful for their contribution to improving our work and advancing the scientific discourse in this field. |

Reviewer 2 Report
Comments and Suggestions for Authors
Iribarnegaray et al., have submitted the manuscript entitled "Droplet digital PCR enhances sensitivity and accuracy in canine distemper virus detection" which is well presented and drafted.
The methodology, result and discussion is satisfactory.
Please following to make manuscript well presented-
In table no. 3 (or in new table), please present the sample type wise (blood, urine etc.) 'true positive' and 'true negative' (seperately) result. So that it will be more clear through table in regards to different sample type.
Comments on the Quality of English Languageminor
Author Response
Comment 1: In table no. 3 (or in new table), please present the sample type wise (blood, urine etc.) 'true positive' and 'true negative' (seperately) result. So that it will be more clear through table in regards to different sample type.
Response 1: We would like to express our sincere gratitude for the time and effort you dedicated to the thorough review of our manuscript. Your comments and suggestions have been immensely valuable and have significantly contributed to improving the quality and clarity of our work. We have modified table N°3 by attaching a new version of the article.
Reviewer 3 Report
Comments and Suggestions for Authors
In this study, authors compared the diagnostic efficacy of ddPCR with RT-PCR and RT-qPCR. They underscored ddPCR's superior sensitivity and concordance with clinical CDV diagnosis.
1. Is there currently a golden standard for the detection of CDV RNA? How to confirm the CDV RNA positive and negative samples? Also, how to calculate TPV and TNV if these samples cannot be confirmed to be CDV RNA positive or negative?
2. The quality of Figure 2B, Figure 5A and Figure 5B needs to be improved.
3. Figure 2B: Please label the dilutions in the figure.
4. Figure 5: Can authors specify the viral loads at least in the figure legend? How to define “high viral loads” and “low viral loads”? Better to label in the figure as well.
Author Response
|
Comments 1: 1. Is there currently a golden standard for the detection of CDV RNA? How to confirm the CDV RNA positive and negative samples? Also, how to calculate TPV and TNV if these samples cannot be confirmed to be CDV RNA positive or negative?
|
|
Response 1: We appreciate the reviewer's insightful questions regarding the gold standard for CDV RNA detection and sample confirmation. Currently, there is no universally accepted gold standard for CDV RNA detection. RT-qPCR is generally considered the most sensitive and specific method available in clinical practice, but it's not infallible. Definitive confirmation of CDV RNA positive and negative samples would ideally involve a combination of methods, including RT-PCR followed by sequencing, virus isolation in cell culture, and immunohistochemistry on tissue samples (for post-mortem cases). However, these comprehensive confirmatory tests are often impractical in large-scale clinical studies due to time and resource constraints. In our study, we used clinical diagnosis by experienced veterinarians as the reference for calculating TPV and TNV. We acknowledge this is not ideal, but it reflects real-world clinical scenarios where decisions often must be made based on clinical presentation. This approach is common in field studies where exhaustive confirmatory testing of all samples is not feasible.
To address this limitation, we have added a paragraph in the Discussion section acknowledging this as a potential limitation of our study and suggesting that future research could benefit from more comprehensive confirmatory testing on a subset of samples. The added paragraph in line 432-453.
|
|
Comments 2: The quality of Figure 2B, Figure 5A and Figure 5B needs to be improved. |
|
Response 2: We appreciate the reviewer's suggestion and we have, accordingly, changed the quality to improve this point.
Comments 3: Figure 2B: Please label the dilutions in the figure. Response 3: Thank you very much for the comment, we agree with it. To improve the figure we include the dilution in the figure.
Comments 4: Figure 5: Can authors specify the viral loads at least in the figure legend? How to define “high viral loads” and “low viral loads”? Better to label in the figure as well. Response 4: Thank you for pointing this out. We agree with this comment. Therefore, we have modified the figure to include the value of each sample in the legend.
|
|
4. Response to Comments on the Quality of English Language |
|
Point 1: |
|
Response 1: (in red) |
|
5. Additional clarifications |
|
We would like to express our sincere gratitude to the reviewers for their thorough and insightful comments. Their expertise and attention to detail have significantly enhanced the quality and clarity of our manuscript. Following the reviewers' suggestions, we have made comprehensive revisions throughout the paper. These changes have improved the precision of our terminology, the clarity of our methodology, and the presentation of our results. We believe that these revisions have strengthened our manuscript, making it more accessible and valuable to the readership of VIRUSES. The process of addressing the reviewers' comments has also led us to refine our analysis and sharpen our conclusions. We are confident that the revised manuscript now presents a more robust and nuanced examination of droplet digital PCR in canine distemper virus detection. We appreciate the opportunity to contribute to the field of virology through VIRUSES. We believe that our study, with these revisions, offers important insights into molecular diagnostic techniques for CDV, which may have broader implications for viral detection methodologies. Once again, we thank the reviewers and editors for their time and expertise. We are grateful for their contribution to improving our work and advancing the scientific discourse in this field. |

Round 2
Reviewer 1 Report
Comments and Suggestions for Authors
The authors have addressed all of my comments and accepted all of my suggestions.
Reviewer 3 Report
Comments and Suggestions for Authors
Authors have addressed all questions.